# Tough Love Lessons: Lateral Violence among Hospital Nurses

**DOI:** 10.3390/ijerph18179183

**Published:** 2021-08-31

**Authors:** María Joao Vidal-Alves, David Pina, Esteban Puente-López, Aurelio Luna-Maldonado, Aurelio Luna Ruiz-Cabello, Teresa Magalhães, Yolanda Pina-López, José Antonio Ruiz-Hernández, Begoña Martínez Jarreta

**Affiliations:** 1Department of Socio-sanitary Sciences, University of Murcia, 30100 Murcia, Spain; mariajoaoalves.acj@gmail.com (M.J.V.-A.); aurluna@um.es (A.L.-M.); aurelio.luna@um.es (A.L.R.-C.); 2Department of Community Medicine, Information and Health Decisions, School of Medicine, University of Porto, 4200-319 Porto, Portugal; imlfmup@gmail.com; 3Department of Science, University Institute of Health Sciences-CESPU, 4585-116 Gandra, Portugal; 4Applied Psychology Service (SEPA), University of Murcia, 30100 Murcia, Spain; esteban.puente@um.es (E.P.-L.); jaruiz@um.es (J.A.R.-H.); 5Department of Nursing, University of Murcia, 30100 Murcia, Spain; yolandapinalopez@hotmail.com; 6Department of Psychiatry and Social Psychology, University of Murcia, 30100 Murcia, Spain; 7Department of Pathological Anatomy, Forensic and Legal Medicine and Toxicology, University of Zaragoza, 50009 Zaragoza, Spain; mjarreta@unizar.es

**Keywords:** lateral violence, nurses, burnout, mental health, job satisfaction

## Abstract

Background: Workplace violence is a growing social problem among many professions, but it particularly affects the health sector. Studies have mainly focused on evaluating user violence toward health professionals, with less attention being paid to other sources of conflict, such as co-workers themselves. There are different manifestations of this violence in what has been called a context of tolerated or normalized violence among co-workers. However, its effects are far from being tolerable, as they have an impact on general health and job satisfaction and contribute to burnout among professionals. Based on this idea, and following the line of the previous literature, nursing staff are a population at high risk of exposure to workplace violence. For this reason, the present study aims to evaluate exposure to lateral violence or violence among co-workers in nursing staff in public health services and the relationship of this exposure with some of the most studied consequences. (2) Methods: A cross-sectional associative study was carried out in which scales of workplace violence (HABS-CS), burnout (MBI-GS), job satisfaction (OJS), and general health (GHQ-28) were applied to a sample of 950 nursing staff from 13 public hospitals located in the southeast of Spain. (3) Results: The results show that nursing staff have a high exposure to violence from their co-workers, which is more common in male nurses. Greater exposure is observed in professionals with between 6 and 10 years of experience in the profession, and it is not characteristic of our sample to receive greater violence when they have less experience or are younger. A positive correlation is observed with high levels of burnout and a negative correlation with general health and job satisfaction. (4) Conclusions: The results of this work contribute to increasing the scientific evidence of the consequences of a type of workplace violence frequent among nursing staff and to which less attention has been paid in relative terms to other types of prevalent violence. Organizations should be aware of the importance of this type of workplace violence, its frequency and impact, and implement appropriate prevention policies that include the promotion of a culture that does not reward violence or minimize reporting. A change of mentality in the academic environment is also recommended in order to promote a more adequate training of nursing staff in this field.

## 1. Introduction

Violence and harassment in the world of work is considered a violation of human rights and a threat to equal opportunities. It is defined as the incidents where professionals are abused, threatened, or assaulted/intimidated in a context related to their work [1]. This violence can be triggered by a set of individual, psychosocial, and work-related factors. Likewise, a workplace culture where abuse is usual and accepted carries the danger of normalizing violence [2,3].

According to the WHO, at least 25% of workplace violence occurs in healthcare settings [4], but, although much has been published on user violence, there is not enough evidence of harassment between co-workers in in this setting yet.

This violence follows different vectors. Vertical violence is the one between a superior and an employee and can be top-down or down-up [5], the former being the most prevalent [6,7]. Lateral violence consists of harassment behaviors between co-workers with equivalent status and consists of physical or verbal violence [5,8]. Lateral violence can also occur in different ways, from person-directed attacks (personal lateral violence) to social isolation (social lateral violence) or work-related (workplace-related lateral violence), the latter being not always seen by the receiver as a form of violence [9].

The few studies on this matter have demonstrated its high prevalence in healthcare and the nursing profession [4,6] and its deleterious effects [10,11], yet several gaps remain to be filled.

### 1.1. Background

Emergency departments generally show higher levels of violence [12,13], but, although the mental health area poses a high risk [14], all health professionals have a high rate of violence and harassment. The difference seems to be that those with greater proximity and/or prolonged exposure to the public are at higher risk, namely physicians [15], but particularly nurses [12,16].

Differences have been found according to individual factors, for instance, personality and gender, but also regarding work-related factors such as job tenure, shifts, type of work, and department (emergency department, primary care) [17,18,19].

Sex remains a topic of discussion since some results point to greater exposure to violence by male professionals [7,16], while others find this variable to have little or no influence [13,20,21], including a meta-analysis of 65 articles [22]. Nonetheless, studies have stated that female professionals are more often a target of psychological harassment behaviors [6] and males suffer more physical violence [15]. Age is also a factor of vulnerability since younger professionals are more often harassed [5] while personality may also contribute (e.g., negative affectivity) to incivility and worker-to-worker violence [23,24].

Psychological violence among co-workers seems to be more frequent among shift workers than among fixed schedule workers [25] with differences between shifts [26].

A 10-European-Country-study in 2008 with 39,898 nursing professionals found that harassment by superiors is very frequent (21%) and that male nurses, younger nurses, and nursing aides are at higher risk of violence compared to female nurses, older nurses, and registered nurses [11].

However, violence between co-workers has been scarcely studied compared to other types of workplace violence. Currently, the problem is far from solved, since the 2015 Eurofound European Working Conditions found that 17% of all male and female participants were victims of workplace harassment and 7% of all participants suffered some sort of exclusion or discrimination, with an increasing tendency, considering previous rates (2005—5%; 2010—6%) [4].

In 2020, due to the pandemic, an overall change of working habits led to more people working from home, but physical exhaustion at the end of the day was more often reported by young and female respondents (35% of women), according to the 2020 Eurofound report. Additional challenges appeared during these difficult times, such as feeling more at risk of contracting COVID-19 due to their jobs (nearly 80% of employees), being highly exposed to emotionally demanding situations (referred by more than 30% in France, Lithuania, Portugal, and Spain), as well as facing the lack of personal protective equipment (PPE), as reported by 3 out of 10 employees [27]. This is common ground for health professionals, who are referred to as the most likely to have higher levels of emotional demands in this report. However, the report did not evaluate workplace violence so it is unknown how the recent changes in the workplace experienced by health professionals have affected this phenomenon.

It is important to note that these figures are just the tip of the iceberg, meaning that not all incidents are reported, either due to the lack of knowledge, fear, or any other reason [28].

There is evidence that nearly 60% of new nurses leave their job due to some form of verbal abuse from a co-worker in their first 6 months of work [29] and that violence is generally strongly related to the intention to leave the nursing profession [30], change institution, and burnout [11].

The reasons found in the literature for the greater vulnerability of nurses to lateral violence is the proximity and constant interaction with patients and working in coordinated teams [31]. Besides, nurses can be at higher risk of violence from co-workers due to being new to the job, having been promoted (when others find it unfair), having relational difficulties, receiving special attention from physicians or working in understaffing conditions [32]. Occupational hazards that come from this close interaction vary according to services, individual characteristics, and work-related issues. A European study found that fixed night shift workers are at a higher risk of violence, which is associated with a higher incidence of burnout, while working part-time seems to be associated with less violent events [11].

A latent class cluster analysis by Einarsen et al. [33] points to the importance of identifying target groups more prone to suffering behaviors that range from incivility to aggression. The former, defined as “rude and discourteous actions of gossiping and spreading rumors, and refusing to assist a co-worker” [34], is more frequent, but the latter, seen as “repeated, unwanted harmful actions intended to humiliate, offend, and cause distress in the recipient”, is characterized by its severity and reiteration [34].

In a study about type-III violence (worker-to-worker violence), with 185 perpetrators, all health professionals, most were female (74.4%), 60% worked full-time, had a mean age of 45.2 years, and a mean job tenure of 11.7 years in the hospital. Nurses are common victims of violence but also common perpetrators [35]. The origins of this violence have been related to the patriarchal historical background of nursing, with its internalized sexism, due to the oppression that it generates, both individually and collectively [1]. This is supported by a more recent content analysis of nurses in California, which states that lateral violence may be used as a form of informal power as a result of organization-related feelings of oppression, meaning that those who feel oppressed (or undervalued) may try to regain power by hurting their colleagues [2].

The consequences of violent incidents against health professionals are severe and include (1) waste of time, (2) mental health problems (PTSD, depression, burnout, depersonalization), (3) lack of sense of safety, and (4) emotional distress (anger, humiliation, fear, guilt) [15].

Given the high prevalence of this type of violence, it is important to note its short- and long-term repercussions, the latter being the most frequently observed. Victims often feel humiliated and undervalued, which affects their relationship with the work environment. It also affects their relationships, causing physical and emotional fatigue, and depressive, anxiety, or stress symptoms [36,37,38]. Specifically, lateral violence between nurses further results in low self-esteem, feelings of powerlessness, and negative patient outcomes [39], besides contributing to burnout [40] and job dissatisfaction [13,41].

A wide range of mental health consequences is expected from this phenomenon, such as anxiety, vulnerability, guilt, anger, sadness, and peer blaming following violence exposures [42], as well as fear, shame, helplessness, chronic fatigue, depression, sleep disorders [43], and, in severe cases, post-traumatic stress syndrome (PTSD) and increased suicide risk [10]. Physical disorders such as musculoskeletal disorders and a heightened risk of cardiovascular disease have been reported [44,45].

Harassment in the workplace has been proven to cause negative repercussions on job satisfaction to an extent that the more situations workers faced, the less satisfied they were [10,46] affecting the quality of care [30,47]. Additionally, the quality of teamwork seems to be part of/fuel/support the occurrence of violence when reduced or lacking, leading to burnout symptoms [11,48]. Lateral violence may be triggered by the enmity or animosity between co-workers that gradually turns into persistent and long-lasting harassment and is often perceived as a turnover from a previous relationship of friendship and trust. It is also a group phenomenon toward one individual and causes severe anxiety to the victim [49].

The main hypothesis stated by the present study is that the perceived violence in this setting by the nursing staff of hospitals by their co-workers (lateral violence) is related to distinct sociodemographic and work variables also found in literature on user violence. It is expected that differences are found concerning sex, age, area of work, job tenure, and length of time in the job. It is further anticipated that the consequences associated with workplace violence most commonly found in the literature (burnout, job satisfaction, and health problems) vary according to the higher or lower perception of lateral violence.

### 1.2. Aims

Since nursing professionals are a population at a high risk of violence, this study addresses the specific individual factors related to the work context.

The specific goals of this study focus on the lateral violence perceived by nursing professionals in public hospitals in the Region of Murcia, Spain, as follows: (1) identifying differences associated with higher exposure to personal-, social-, and work-related violence according to socio-demographic and socio-occupational variables; (2) empirically obtaining subgroups depending on their exposure to violence, and analyzing them according to their reported levels of burnout, satisfaction, and general health.

## 2. Materials and Methods

This is a cross-sectional associative study using self-report questionnaires. The target population consisted of nursing professionals from public hospitals in the southeast of Spain.

### 2.1. Sample

A random block sampling was performed, prompting a total sample of 950 nursing professionals from 13 public hospitals located in the southeast of Spain, 6 of which were considered large (with a bed capacity of more than 200 beds) and 7 medium or small (with a bed capacity of 200 beds or less).

Considering the characteristics of the sample (Table 1), the age range of participants was from 30 to 50 years, with a mean age of 39.43 years (SD = 9.65). Most were women (77.8%) and were married or cohabiting (63.2%). Concerning work characteristics, 54.3% were in the nursing profession for 0 to 5 years (with a mean of 14.02 years) and at least 54% were in the same job position for the last 5 years (mean 7.31 years, SD = 8.35). From the studied sample, 20.3% worked in surgery, 17% in internal medicine, 14.3% in the emergency department, 6.9% in day care, 5.5% in mental health, and 14.8% in other facilities.

### 2.2. Instrument

A 76-item protocol including the above-mentioned sociodemographic variables (such as age, sex, marital status) and work-related variables (length of time in the profession, hospital, length of time in the current job position, and type of unit) was used. Other scales were used to measure lateral workplace violence, burnout, job satisfaction, and general health.

#### 2.2.1. Health Workers’ Aggressive Behavior Scale–Co-Workers and Superiors (HABS-CS)

The HABS-CS was created by Waschgler et al. [50] and assesses hostility behaviors perceived by health professionals by co-workers. It encompasses 10 items with 6 response options (1 = never to 6 = daily) grouped into three factors: personal factors, social factors, and work-related factors. The original study presents an internal consistency of 0.82 for the personal scale, 0.79 to social scale, and 0.72 to work-related scale, with a total Cronbach Alfa of 0.864. For the present study, the total reliability of 0.87 was met for the total scale and 0.82, 0.82, and 0.71 in the personal, social, and work factors, respectively.

#### 2.2.2. Maslach Burnout Inventory-General Survey (MBI-GS)

Created by Schaufeli et al. (1996), this study used the Spanish version, translated and validated by Gil–Monte [51]. It includes 16 items grouped in three dimensions: emotional exhaustion, professional efficacy, and cynicism, with 5, 5, and 6 items, respectively. Responses range from 0 (never) to 6 (always). Gil–Monte’s study presents an internal consistency of 0.83 for emotional exhaustion, 0.72 for professional efficacy, and 0.73 for cynicism [51]. To the present study, internal consistency of 0.85 was found for emotional exhaustion, 0.70 to cynicism, and 0.85 to professional efficacy.

#### 2.2.3. Overall Job Satisfaction (OJS)

This scale, first built by Warr, Cook, and Wall [52], was adapted to Spanish by Pérez and Fildalgo [53], the version used in this study. It encompasses 15 items organized in two subscales: intrinsic satisfaction (factors related to responsibility and work recognition) and extrinsic satisfaction (organizational work-related factors such as work schedule). Responses range from 1 (very dissatisfied) to 7 (very satisfied). The original study presented an internal consistency of 0.85 to 0.88 for extrinsic factors and 0.74 to 0.78 for intrinsic factors. The present study yields 0.84 for the former and 0.70 for the latter.

#### 2.2.4. General Health Questionnaire (GHQ-28)

Proposed by Goldberg and Hillier (1979), this scale measures general health. Its Spanish version, used in the present study, was adapted by Lobo, Pérez–Echevarría, and Antral [54], including 28 items grouped into 4 subscales: psychological somatic symptoms (somatic GHQ), anxiety and insomnia (anxiety GHQ), social dysfunction scale (dysfunction GHQ), and depressive symptoms scale (depression GHQ). Responses are provided in four options from zero to three (0–3), going from lower to higher intensity. The original study’s internal consistency was 0.78 for somatic GHQ, 0.85 for anxiety GHQ, 0.75 for dysfunction GHQ, and 0.78 for depression GHQ. The present study yielded 0.79 for both somatic GHQ and anxiety GHQ, 0.71 for dysfunction GHQ, and 0.78 for depression GHQ.

### 2.3. Procedure

For sampling purposes, the authors contacted the directors and supervisors of the participating hospitals to provide them with detailed information on the present study and its goals. Upon acceptance, a meeting was arranged with the supervisors of the different units (as fellow researchers) during which the study protocols were delivered. These included an informative note, the above-mentioned scales, and instructions regarding its completion, informed consent, and delivery to the research team in a sealed envelope. A code was ascribed to each worker and protocols were randomly assigned to 50% of the sample. The protocol was delivered by fellow researchers who later managed its reception, in a sealed envelope without identification in a maximum deadline of two weeks. Protocols undelivered during this time length were considered lost. A response rate of 70.48% was obtained.

The present study, designed under STROBE guidelines, was approved by the Ethics Committee of the University of Murcia and the board of directors of each hospital. The authors declare no conflict of interest.

### 2.4. Data Analysis

Data analysis for the present study was performed using SPSS version 25. The sample distribution (mean and standard deviation) and response percentages according to the study variables of the ad-hoc questionnaire were analyzed. A Student’s *t*-test was used to compare the mean of dichotomic variables and ANOVA for the analysis of variance of the factors with more than two levels. Tukey’s test for post hoc analysis was used to delve into such differences. For this purpose, the size effect was estimated as well as the Cohen’s d for mean differences, partial Eta squared (η2) for variance differences, and r for post hoc analysis. The Pearson correlation test was used to complete the analysis of the relationship between the scales used.

## 3. Results

It was possible to observe that at least 59.2% of the sample was exposed to violence from a co-worker at least once in the last year. Specifically, 51% of the sample was exposed to lateral violence of a personal nature (e.g., “Some co-workers spread false rumors about me”), 37.3% of social nature (e.g., “Some co-workers have stopped talking to me”), and 21.3% work-related (e.g., “Some co-workers deliberately accuse me of other people’s mistakes”).

### 3.1. Personal Lateral Violence

Co-worker violence of a personal nature displayed significant differences according to the sex of the respondent, being higher in male participants (t = 2.16 *p* = 0.03 d = 0.17). Further, workers within the age range of 30–50 years seemed to be at higher risk, followed by those younger than 30 and those 50 or older (F = 3.23 *p* = 0.02 d = 0.01). Concerning time in the profession, those with 6 to 11 years in the profession perceived more violence from co-workers, followed by those with 12 to 20 years, less than 5, and more than 20 (F = 2.61 *p* = 0.03 d = 0.01). The time in the job position yielded significant differences, its influence being higher in the 11-to-15-year interval, followed by 6 to 10, 1 to 5, more than 15, and less than a year (F = 2.90 *p* = 0.01 d = 0.02). All differences found revealed a low effect size (Table 2 and Table 3).

### 3.2. Social Lateral Violence

No significant differences were found concerning sex and type of hospital (Table 4). The figures obtained for this factor, though, yielded significant differences related to the type of unit, although again with a low effect size, showing a higher prevalence in external consultations and outpatient units (F = 2.19 *p* = 0.04 d = 0.02) (Table 5). No significant differences were found concerning the other variables studied (Table 4 and Table 5).

### 3.3. Work-Related Lateral Violence

Concerning work-related violence between co-workers, sex of the respondent and type of hospital did not present significant differences (Table 6). On the other hand, significant differences were found according to the type of unit (Table 7). The units with the highest exposure to this violence were outpatient and external consultation, followed by surgery, other, mental health, maternal and child care, emergency department, and internal medicine (F = 2.8 *p* = 0.01 d = 0.02) (Table 7).

### 3.4. Lateral Violence and External Correlates

Concerning the relationship between social lateral violence and possible consequences for health professionals, Pearson correlations were obtained confirming that personal lateral violence is significantly negatively correlated to both extrinsic (r = −0.18, *p* = 0.01) and intrinsic satisfaction (r = −0.19, *p* = 0.01). On the other hand, a significant positive correlation was found between personal lateral violence and emotional exhaustion (r = 0.28, *p* = 0.01), cynicism (r = 0.21, *p* = 0.01), somatic symptoms (r = 0.21, *p* = 0.01), anxiety (r = 0.24, *p* = 0.01), social dysfunction (r = 0.07, *p* = 0.05), and depression (r = 0.20 *p* = 0.01). No significant correlation was found with perceived professional efficacy.

Referring to the social type of lateral violence, a significant negative correlation emerged with satisfaction in both its scopes, with r values of 0.18 (*p* = 0.01) for extrinsic satisfaction and 0.20 (*p* = 0.01) for intrinsic satisfaction. A significant positive correlation with emotional exhaustion (r = 0.20, =.01), cynicism (r = 0.18, *p* = 0.01), and the variables measured by GHQ-28: somatic symptoms (r = 0.17, *p* = 0.01), anxiety (r = 0.16, *p* = 0.01), social dysfunction (r = 0.12, *p* = 0.01), and depression (r = 0.20, *p* = 0.01) was encountered. No relationship was found to professional efficacy (r = −0.05, *p* = 0.05).

Lastly, when studying the relationship between the work-related lateral violence and the hypothesized consequences, a negative correlation was found with both extrinsic (r = −0.17, *p* = 0.01) and intrinsic satisfaction (r = −0.22, *p* = 0.01) (Table 8). On the other hand, positive correlations were found with emotional exhaustion, cynicism (r = 0.17, *p* = 0.01 for both), anxiety and social dysfunction (r = 0.16, *p* = 0.01), somatic symptoms (r = 0.15, *p* = 0.01), and depression (r = 0.18, *p* = 0.01). However, no correlation was found with professional efficacy (r = −0.04, *p* = 0.05) (Table 8).

## 4. Discussion

The goals of the present study were to measure the lateral violence perceived by nursing professionals and identify differences associated with higher exposure to personal, social and work-related violence according to sociodemographic and socio-occupational variables. Additionally, an analysis according to the reported levels of burnout, job satisfaction, and general health was also envisioned.

The results of the present study sustain that nursing professionals are highly exposed to co-worker violence, in line with other studies [6,11,33,35,55].

This violence is perceived by a high percentage of nurses in their workplace as disruptive and inappropriate behavior by one employee toward another, whether in an equal or inferior position. This is partly due to their demanding and highly supervised environment [56,57]. This violent experience causes stress and negatively impacts psychological health [5,58], leads nurses to consider leaving their jobs [58,59,60,61], or adversely impacts patient care and attention if they stay in the job [62].

### 4.1. Sex

The observed differences in personal lateral violence concerning sex are in line with the literature, which identifies male nurses as more at risk of workplace violence [63,64] and, more specifically, verbal violence [65].

On the other hand, some authors place female nurses at higher risk depending on the hospital setting [66] and find cultural differences in exposure to different types of violence (e.g., more physical violence against men in cultures where women are usually seen as frail and in need of protection) [55]. Others found no significant sex differences in lateral/horizontal violence [67].

### 4.2. Age, Time in the Profession and Time in the Job

In the present sample, with a mean of 14.02 years in the nursing profession, the most affected by personal lateral violence were those with 6 to 10 years in the profession, being below average. A qualitative approach sustained that perpetrators target those newer to the job and younger because they are easy targets and, being less experienced, they are less likely to be able to defend themselves [60], consistent with the tendency found by other research [58]. In addition, acts of a personal nature such as gossiping, boycotting opportunities, tough love, or sink-or-swim are common incivilities in young nurses’ education, in the context of a permissive culture of vertical or lateral violence [60], but can also be found in nurses who are not necessarily young, but younger than their leaders [68].

In the present study, it was not the youngest nurses who perceived more workplace violence which, in general, challenges studies that indicate a higher risk in younger people of experiencing more violence among co-workers [69]. The differences found between the present study and those found in the literature require further research to check if they are a typical characteristic of this population or due to another feature.

A mixed-method approach suggests that, when younger nurses work together with older ones, the former is more proactive and straightforward and the latter tend to be more conservative, which is a conflict generator [70]. As we see it, lateral violence in healthcare settings is strongly a nurse-to-nurse phenomenon that is frequently based on institutionalized tolerance to it and to the idea that hierarchy needs this type of behavior to maintain civility in the workplace [59,70]. It also lays in a culture of legitimized worker-to-worker violence in healthcare [32] and is frequently found in Latin European countries, keeping a relationship with low power distance [71].

Norton et al. [6] found that nurses suffer more vertical than horizontal violence (74.2% vs. 25.8%, *p* < 0.029) which confirms the normalization of the use of violence in healthcare by people in a higher position to control those in a lower one. Despite the current results, the term formulated by Meissner that “nurses eat their young” does not seem out of place when we face this top-down violence tendency used as a mean of authority [72].

Time in the profession and the job position also differs from data found in the literature on the exposure to workplace violence, since those with medium time were the ones most affected by peer violence (neither the newest in the profession or job position nor those with more time in both). Although these variables have a low effect size, it is possible to contextualize that, of the age range of the participants in the present study, the age group most prone to be victimized during training [68] was not represented here.

A study using the NAQ-R states that co-worker violence such as withholding information that affects performance, having opinions ignored, or being forced to work below the individual’s competence is often identified by nurses with an average time in the profession of 20 years [73]. Dellasega [74] points out that, besides nurses under training, newly hired nurses with experience, independently of age, are also often targeted by co-workers. On the other hand, the present data point that older nurses (aged more than 50) are less likely to get harassed, which can be related to perpetration being more prevalent among nurse leaders and staff toward younger nurses [68]. Additionally, the risk of co-worker violence was found to decline as nurses’ length of service and age increased in a cluster analysis by Karatuna et al. [71].

### 4.3. Shift or Fixed Schedule and Setting

Concerning higher perceived co-worker violence in the case of shift work, the present results are corroborated by previous studies. Shift-working nurses have been proven to be at higher risk of vertical and lateral violence than fixed schedule workers [6,75].

The results for social lateral violence factor point to a higher prevalence in external consultations and outpatient units although without significant differences, although there is reference in the literature to greater impact of lateral violence in the Emergency Room (ER) [55,76].

### 4.4. Burnout, Job Satisfaction, and General Health

As expected, personal-, social-, and work-related lateral violence are significantly negatively correlated to both extrinsic and intrinsic satisfaction and positively correlated to dimensions of burnout and poorer health quality, as happens in other types of workplace violence, such as user violence [12,19,77]. The present results confirm other data found for burnout, namely positioning co-worker violence as a predictive factor for burnout (β = 0.37 *p* < 0.001) and holding a negative correlation with job efficiency (r = −0 322, *p* < 0.01) [78]. This may be related to the interference of violence in workers’ wellbeing, representing an additional source of stress, especially in the long run, when it becomes toxic (known to be health-disruptive) and negatively impacts self-regulating body functions and psychological health [3].

In an ER-based study, 91.7% of respondents stated that lateral violence decreases their job satisfaction, with 53.3% pondering transferring to another unit or hospital, or leaving their job [79]. Absenteeism has also been shown to be 1.5 times higher in comparison to non-victimized peers (95% CI: 1.3–1.7) and the intention to leave rises to 78.5% of bullied nurses with a length of service lower than 5 years [78].

This study is not without limitations, so the data reported in it should be interpreted with caution. The cross-sectional design does not allow us to make causal relationships, limiting us to the description and comparison of the data. The differences found between our study and previous studies may be due to specific circumstances that were not taken into account, so it would be interesting to propose designs that allow us to study them in depth. Furthermore, although measures were taken to encourage response and anonymity, self-reported questionnaires can be another source of bias, especially if they collect sensitive information that may affect the worker’s work environment. Finally, although the sample size is large, the sample belongs to a specific region, which may lead to differences with other populations both nationally and internationally.

## 5. Conclusions

Experience of personal lateral violence in nurses is strongly positively correlated to higher levels of burnout and poorer psychological health indicators. This type of workplace violence also negatively impacts both extrinsic and intrinsic job satisfaction.

The culture of fostering peer violence in nursing exists when organizations allow, ignore, or reward such behaviors and leaders minimize complaints, which suggest that combating this phenomenon requires organizational support and bystander empowerment policies [59]. This supports the common idea that nurses “eat their young”, a mindset that is also enabled by nursing academic training [72].

We highlight that victimized nurses may not report lateral violence out of fear of retaliation or because they see it as necessary for nursing education, encoded in a “sink or swim” mindset. Similarly, other nurses and health professionals may, as bystanders, witness this phenomenon but lack skills on what to do or fear being the next in line to be harassed [80].

In general terms, intervention, prevention, training, or support programs for professionals are focused on user violence. Although this phenomenon is of special interest, in our opinion, both the approach to these proposals and their implementation must have qualitative and quantitative differences when the objective is to reduce violence among co-workers. For this reason, our study provides evidence of this reality and facilitates the planning of specific programs aimed at this objective which, as has been observed, may be affecting both the health of professionals and their work performance. For subsequent studies, it would be interesting to know the biopsychosocial profile of these professionals, allowing the design of even more specific programs. It is also advisable to explore which of the variables studied for the reduction of violence by users against healthcare personnel are effective for the reduction of violence among co-workers. Finally, longitudinal studies grouping these two sides of the same reality could serve as a basis for improving the work environment of healthcare professionals.

## Figures and Tables

**Table 1 ijerph-18-09183-t001:** Sociodemographic and work-related variables.

Variables	*n*	%
**Sex**		
Female	720	77.8
Male	199	21.5
Missing data	6	0.6
**Marital status**		
Single	290	31.4
Married or cohabiting	585	63.2
Divorced, separated and/or widow	45	4.9
Missing data	5	0.5
**Age**		
<30	137	14.8
30–50	586	63.4
+50	153	16.5
Missing data	49	5.3
**Length of time in the profession (years)**		
<5	502	54.3
6–11	208	22.5
12–20	120	13.0
>20	70	7.6
Missing data	25	2.7
**Length of time in the job position (years)**		
<1	120	13.0
1–5	382	41.3
6–10	180	19.5
11–15	84	9.1
>15	134	14.5
Missing data	25	2.7
**Units**		
Surgery	188	20.3
Maternal and child care	101	10.9
Internal medicine	161	17.4
Emergency department	132	14.3
External consultations/outpatient	64	6.9
Mental health	51	5.5
Other	137	14.8
Missing data	91	9.8
**Type of hospital**		
Large	761	82.3
Medium or small	164	17.7

**Table 2 ijerph-18-09183-t002:** *T*-test for personal lateral violence.

Variables	Mean (DT)	t	*p*	IC 95%		d
				Inf.	Sup.	
**Sex**						
Female	5.90 (3.06)	2.16	0.03	0.05	1.05	0.17
Male	6.46 (3.41)					
**Type of hospital**						
Large	6.03 (3.14)	0.30	0.89	−0.45	0.61	0.02
Medium or small	5.95 (3.12)					

**Table 3 ijerph-18-09183-t003:** ANOVA for personal lateral violence.

Variables	Media (DT)	F	*p*	η2
**Marital status**				
Single	6.30 (3.22)	1.77	0.17	0.01
Married or cohabiting	5.90 (3.15)			
Divorced, separated, or widow	5.67 (2.40)			
**Age**				
<30	5.81 (2.80)	3.23	0.02	0.01
30–50	6.16 (3.22)			
+50	5.39 (2.78)			
**Length of time in the profession (years)**				
<5	5.95 (3.06)	2.61	0.03	0.01
6–11	6.48 (3.48)			
12–20	6.06 (3.31)			
>20	5.24 (2.47)			
**Length of time in the job position (years)**				
<1	5.48 (2.50)	2.90	0.01	0.02
1–5	6.10 (3.20)			
6–10	6.41 (3.41)			
11–15	6.57 (3.50)			
>15	5.49 (2.90)			
**Units**				
Surgery	6.14 (3.18)	0.86	0.52	0.01
Maternal and child care	6.51 (3.98)			
Internal medicine	5.88 (2.94)			
Emergencies	6.08 (2.75)			
External consultation/outpatient	6.40 (4.18)			
Mental health	5.88 (2.78)			
Other	5.69 (2.93)			

**Table 4 ijerph-18-09183-t004:** Student’s *t*-test for social lateral violence.

Variables	Mean (DT)	t	*p*	IC 95%		d
				Inf.	Sup.	
**Sex**						
Female	3.75 (1.92)	0.62	0.53	−0.20	0.38	0.06
Male	3.85 (1.57)					
**Type of hospital**						
Large	3.74 (1.85)	−0.96	0.34	−0.47	0.16	0.08
Medium or small	3.89 (1.80)					

**Table 5 ijerph-18-09183-t005:** ANOVA for social lateral violence.

Variables	Media (DT)	F	*p*	η2
**Marital status**				
Single	3.75 (1.59)	0.52	0.59	0.01
Married or cohabiting	3.80 (2.01)			
Divorced, separated, or widow	3.52 (1.04)			
**Age**				
<30	3.58 (1.44)	1.71	0.16	0.01
30–50	3.83 (1.80)			
+50	3.60 (2.10)			
**Length of time in the profession (years)**				
<5	3.77 (1.84)	0.78	0.54	0.01
6–11	3.89 (2.02)			
12–20	3.79 (1.84)			
>20	3.48 (1.54)			
**Length of time in the job position (years)**				
<1	3.82 (2.02)	1.50	0.19	0.01
1–5	3.75 (1.78)			
6–10	3.79 (1.88)			
11–15	4.20 (2.37)			
>15	3.53 (1.48)			
**Units**				
Surgery	3.95 (2.01)	2.19	0.04	0.02
Maternal and child care	3.66 (1.50)			
Internal medicine	3.42 (1.06)			
Emergency department	3.82 (1.58)			
External consultation/outpatient	4.30 (3.08)			
Mental health	3.74 (1.86)			
Other	3.75 (2.01)			

**Table 6 ijerph-18-09183-t006:** *T*-test for work-related lateral violence.

Variables	Mean (DT)	t	*p*	IC 95%		d
				Inf.	Sup.	
**Sex**						
Female	3.51 (1.51)	0.074	0.94	−0.22	0.24	0.01
Male	3.52 (1.28)					
**Type of hospital**						
Large	3.51 (1.47)	0.17	0.99	−0.24	0.24	-
Medium or small	3.51 (1.42)					

**Table 7 ijerph-18-09183-t007:** ANOVA for work-related lateral violence.

Variables	Mean (DT)	ANOVA	*p*	η2
**Marital status**				
Single	3.47 (1.28)	0.23	0.79	0.01
Married or cohabiting	3.54 (1.54)			
Divorced, separated, or widow	3.49 (1.55)			
**Age**				
<30	3.40 (0.85)	0.75	0.52	0.01
30–50	3.50 (1.39)			
+50	3.57 (1.95)			
**Length of time in the profession (years)**				
<5	3.54 (1.43)	0.71	0.59	0.01
6–11	3.55 (1.70)			
12–20	3.30 (0.92)			
>20	3.56 (1.74)			
**Length of time in the job position (years)**				
<1	3.52 (1.11)	0.22	0.96	0.01
1–5	3.55 (1.52)			
6–10	3.52 (1.65)			
11–15	3.51 (1.50)			
>15	3.39 (1.35)			
**Units**				
Surgery	3.65 (1.34)	2.8	0.01	0.02
Maternal and child care	3.41 (1.25)			
Internal medicine	3.29 (0.79)			
Emergency department	3.31 (0.77)			
External consultation/outpatient	4.01 (2.57)			
Mental health	3.51 (1.79)			
Other	3.64 (1.91)			

**Table 8 ijerph-18-09183-t008:** Lateral violence consequences.

	1	2	3
**1.** HABS personal	1		
**2.** HABS social	0.55 **	1	
**3.** HABS work-related	0.50 **	0.57 **	1
**4.** Extrinsic satisfaction	−0.18 **	−0.18 **	−0.17 **
**5.** Intrinsic satisfaction	−0.19 **	−0.20 **	−0.22 **
**6.** Emotional exhaustion	0.28 **	0.20 **	0.17 **
**7.** Professional efficacy	−0.04	−0.05	−0.04
**8.** Cynicism	0.21 **	0.18 **	0.17 **
**9.** Somatic symptoms	0.21 **	0.17 **	0.15 **
**10.** Anxiety	0.24 **	0.16 **	0.16 **
**11.** Social dysfunction	0.07 *	0.12 **	0.16 **
**12.** Depression	0.20 **	0.20 **	0.18 **

* = *p* < 0.05; ** = *p* < 0.01.

## Data Availability

The data presented in this study are available on request from the corresponding author. The data are not publicly available due to the confidentiality agreement with the participating organizations.

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
