# Peer review of "Tough Love Lessons: Lateral Violence among Hospital Nurses"

_ijerph, 2021, doi:10.3390/ijerph18179183_

Round 1

Reviewer 1 Report

The topic in this manuscript is of importance, which is timely at the moment. The methods used are described in the manuscript. The style of the manuscript is also structured and easy to follow. However, there are some comments I need to emphasize as below.

Results

Please mark tables 5 to 8 to where they are appropriate (e.g. Put "Table 5" at the end of the sentence where the results were explained) in the results section of the manuscript. And then explain each of them in more details so that readers can interpret the meanings of the results.

Discussion

Please interpret the meaningful results by comparing previous research. Researchers need to analyze and interpret what causes those results rather than simply comparing the previous research with the results of this study. For example, in the correlation results of "burnout, job satisfaction, and general health" on page 12, readers would like to know what causes the correlation among these variables deeply. What do the authors think is the reason for these results?

Please describe the limitations of this study in details.

Other matters

Please correct a typo or grammatical errors.

(e.g., Is the use of "liability" in the following sentence in line 203 on page 6, "…, the total liability of .87…" correct?)

Author Response

Dear reviewer, 

We are very grateful for your comments, they have allowed us to improve the quality of our research. 

Below, we specify how we have dealt with each of these comments. 

Please mark tables 5 to 8 to where they are appropriate (e.g. Put "Table 5" at the end of the sentence where the results were explained) in the results section of the manuscript. And then explain each of them in more details so that readers can interpret the meanings of the results.

Thanks for pointing this out. This amendment was done according to the reviewer’s instructions, for which we are grateful.

Discussion

Please interpret the meaningful results by comparing previous research. Researchers need to analyze and interpret what causes those results rather than simply comparing the previous research with the results of this study. For example, in the correlation results of "burnout, job satisfaction, and general health" on page 12, readers would like to know what causes the correlation among these variables deeply. What do the authors think is the reason for these results?

We totally agree with the reviewer’s remark. New information was added to the mentioned paragraph accordingly.

Please describe the limitations of this study in details.

A text pointing out the main limitations of the study has been added. We thank the reviewers for this recommendation.

Other matters

Please correct a typo or grammatical errors.

(e.g., Is the use of "liability" in the following sentence in line 203 on page 6, "…, the total liability of .87…" correct?)

We thank the reviewer for noticing this. It was indeed an error and we changed it to the proper term “reliability”.

Reviewer 2 Report

My only concern with the background of the article is the lack of information reported about historical patriarchal healthcare systems that literature reports, that are oppressive toward nurses.  When nurses are oppressed, they are unable to overcome power structures above them without fear of retaliation (I did read in article some information on retaliation).  If you fear retaliation and you are overworked, understaffed, frustrated, stressed you look to vent these unaddressed oppressions and it leads to moral distress, cynicism, depersonalization, lateral violences, and full burnout syndrome. Therefore, nurses often look laterally to dissipate their frustrations onto their coworkers, because they are nearest to them laterally since they are unable to overcome above oppression.  I think it's time we look not merely within the profession when it comes to lateral violence as the causation, but also external factors that nurses have little control over (top down reasons).  Also, this article was interesting in that it reported that male nurses are subject to more physical violence from coworkers. As a male nurse over two decades I do believe there is great difficulty with advancement in the profession and this is a form of lateral violence as indicated in the text of article and appreciated these remarks.

When reading the first sentence of the abstract: 

"Workplace violence is a growing social problem that, among many other  professionals, particularly affects the health sector."

I think you need to exchange the word "professionals" for "professions" and the sentence will make more sense.   For instance, you could say, "Workplace violence is a growing social problem among many professions, but(or and) particularly affects the health sector."

Author Response

Dear reviewer,

We are very grateful for your comments, they have allowed us to improve the quality of our research.

Below, we specify how we have dealt with each of these comments.

My only concern with the background of the article is the lack of information reported about historical patriarchal healthcare systems that literature reports, that are oppressive toward nurses.  When nurses are oppressed, they are unable to overcome power structures above them without fear of retaliation (I did read in article some information on retaliation). If you fear retaliation and you are overworked, understaffed, frustrated, stressed you look to vent these unaddressed oppressions and it leads to moral distress, cynicism, depersonalization, lateral violences, and full burnout syndrome. Therefore, nurses often look laterally to dissipate their frustrations onto their coworkers, because they are nearest to them laterally since they are unable to overcome above oppression.  

I think it's time we look not merely within the profession when it comes to lateral violence as the causation, but also external factors that nurses have little control over (top down reasons).  Also, this article was interesting in that it reported that male nurses are subject to more physical violence from coworkers. As a male nurse over two decades I do believe there is great difficulty with advancement in the profession and this is a form of lateral violence as indicated in the text of article and appreciated these remarks.

We highly considered this important note from the reviewer. As such, we added information about the background of the nursing profession and its patriarchal structure and internalized sexism, mentioning the work of Purpora et al., 2012. We also mention a more recent content analysis making reference to the use of lateral violence as a source of informal power, resulting from organization-related feelings of oppression (Myers et al., 2016). We also thank the reviewer’s final remark.

When reading the first sentence of the abstract: "Workplace violence is a growing social problem that, among many other  professionals, particularly affects the health sector."

I think you need to exchange the word "professionals" for "professions" and the sentence will make more sense.   For instance, you could say, "Workplace violence is a growing social problem among many professions, but(or and) particularly affects the health sector."

Since we considered the reviewer’s suggestion much clearer, we changed the first sentence of the abstract accordingly.